# Investigating CRISPR/Cas9 gene drive for production of disease-preventing prion gene alleles

**Andrew R. Castle**[1,2], **Serene Wohlgemuth**[1,2], **Luis Arce**[1,3], **David Westaway**[1,2,3]*

**1** Centre for Prions and Protein Folding Diseases, University of Alberta, Edmonton, Alberta, Canada, **2** Department of Medicine, University of Alberta, Edmonton, Alberta, Canada, **3** Department of Biochemistry, University of Alberta, Edmonton, Alberta, Canada

* david.westaway@ualberta.ca

**Data Availability Statement:** All relevant data are within the manuscript and its Supporting Information files.

**Funding:** This study was supported by the Alberta Prion Research Institute (https://albertainnovates.

## Abstract

Prion diseases are a group of fatal neurodegenerative disorders that includes chronic wasting disease, which affects cervids and is highly transmissible. Given that chronic wasting disease prevalence exceeds 30% in some endemic areas of North America, and that eventual transmission to other mammalian species, potentially including humans, cannot be ruled out, novel control strategies beyond population management via hunting and/or culling must be investigated. Prion diseases depend upon post-translational conversion of the cellular prion protein, encoded by the *Prnp* gene, into a disease-associated conformation; ablation of cellular prion protein expression, which is generally well-tolerated, eliminates prion disease susceptibility entirely. Inspired by demonstrations of gene drive in caged mosquito species, we aimed to test whether a CRISPR/Cas9-based gene drive mechanism could, in principle, promote the spread of a null *Prnp* allele among mammalian populations. First, we showed that transient co-expression of Cas9 and *Prnp*-directed guide RNAs in RK13 cells generates indels within the *Prnp* open-reading frame, indicating that repair of Cas9-induced double-strand breaks by non-homologous end-joining had taken place. Second, we integrated a ~1.2 kb donor DNA sequence into the *Prnp* open-reading frame in N2a cells by homology-directed repair following Cas9-induced cleavages and confirmed that integration occurred precisely in most cases. Third, we demonstrated that electroporation of Cas9/guide RNA ribonucleoprotein complexes into fertilised mouse oocytes resulted in pups with a variety of disruptions to the *Prnp* open reading frame, with a new coisogenic line of *Prnp*-null mice obtained as part of this work. However, a technical challenge in obtaining expression of Cas9 in the male germline prevented implementation of a complete gene drive mechanism in mice.

## Introduction

Prion diseases are uniformly fatal, neurodegenerative disorders of mammals. One such disorder is chronic wasting disease (CWD), which affects cervids and is highly transmissible. CWD was first identified in Colorado in the 1960s and, as of 2019, had spread to 26 U.S. states and three Canadian provinces, with cases also detected in several European countries and in South

ca/programs/alberta-prion-research-institute/)
(APRIEP 201600033 and ABIBS 201600023; both
awarded to D.W.) and the Canada Foundation for
Innovation (https://www.innovation.ca/)
(NIF21633; awarded to D.W.). The University of
Alberta Transgenic core RRID:SCR_019175
receives funding from the Faculty of Medicine and
Dentistry, University of Alberta and the Canada
Foundation for Innovation (CFI) awards to
contributing investigators. The University of Alberta
Flow Cytometry core RRID:SCR_019195 receives
funding from Faculty of Medicine and Dentistry, the
LiKa Shing Institute of Virology and the Canada
Foundation for Innovation (CFI) awards to
contributing investigators. The funders had no role
in study design, data collection and analysis,
decision to publish, or preparation of the
manuscript.

**Competing interests:** The authors have declared
that no competing interests exist.

Korea [1]. With disease prevalence in wild cervids exceeding 30% in some endemic areas of North America [2], CWD presents significant challenges for the deer hunting and farming industries. Furthermore, while prion diseases are generally subject to a "species barrier" in terms of transmission [3, 4], this barrier is not absolute, and concerns remain that CWD could spread to other mammalian species [5], perhaps even to humans [6]. Without effective prophylactics or therapeutics, control strategies for CWD are currently restricted, with limited success, to population management via hunting and/or culling [7]. Several vaccination trials have been performed [8–11], with the ultimate aim of protecting farmed cervids, but one of the most recent studies reported that the vaccine candidate actually accelerated CWD onset [11]. It is therefore imperative to seek additional approaches to deal with CWD and other prion diseases that may emerge in animal populations.

The cellular prion protein ($PrP^C$), encoded by the *Prnp* gene, is a glycoprotein that normally resides on the surface of cells [12], particularly neurons [13]. A key hallmark of prion disease is the template-mediated conversion of $PrP^C$ substrate into abnormal, partially protease-resistant conformations often referred to as $PrP^{Sc}$ [14–16]. Thus, partial reduction of $PrP^C$ expression in *Prnp* hemizygotes [17] or use of *Prnp* mRNA-directed antisense oligonucleotides impedes prions infections [18, 19], while complete elimination of $PrP^C$ expression prevents prion infections entirely [20, 21]. Importantly, no severe phenotypes have been detected in genetically engineered $PrP^C$-null animals [22–26] or in goats with a naturally occurring premature stop codon within the *Prnp* open-reading frame (ORF) that ablates $PrP^C$ expression [27]. The only consistently observed phenotype of $PrP^C$-null animals is a relatively late-onset peripheral neuropathy with mild phenotypic consequences [28–30]. Therefore, we asked whether recent advances in gene editing technologies could be exploited to eliminate $PrP^C$ expression and, consequently, impart prion disease resistance into animal populations. Given the absolute requirement for $PrP^C$ substrate, such an approach would have the crucial advantage of conferring resistance to all prion strains unlike methodologies which target misfolded PrP forms. In principle, *Prnp* could be knocked out via gene editing in a small number of captive cervids, which could then be used in breeding programs. However, a gene drive mechanism would theoretically be capable of spreading prion disease resistance much faster and could also be effective in wild cervid populations. Gene drives are "selfish" molecular mechanisms that promote inheritance of a DNA sequence at super-Mendelian frequencies. Such mechanisms exist naturally (reviewed in [31]) and include transposable elements, meiotic drivers, and mating type switching in yeast. Recent years, however, have seen renewed interest in synthetic gene drive systems that take advantage of CRISPR/Cas9 technology. CRISPR/Cas9-based gene drives have proven effective in laboratory settings, particularly in malaria vector mosquito species [32–35], and are being considered seriously for the eradication of invasive species in countries such as New Zealand [36].

Here, in a controlled laboratory setting, we investigated whether a CRISPR/Cas9-based gene drive could, in principle, be used to promote the spread of a null *Prnp* allele. We succeeded in generating reagents able to modify the murine *Prnp* locus in cultured cells via homology-directed repair (HDR) of Cas9-induced double-strand breaks (DSBs) and describe an issue pertaining to germline expression of Cas9 in mice.

## Results

### Design of a CRISPR/Cas9-based gene drive mechanism to spread prion disease resistance

The gene drive mechanism we designed is outlined in Fig 1. Briefly, we planned to create a donor DNA cassette containing Cas9 and guide RNA (gRNA) coding sequences under the

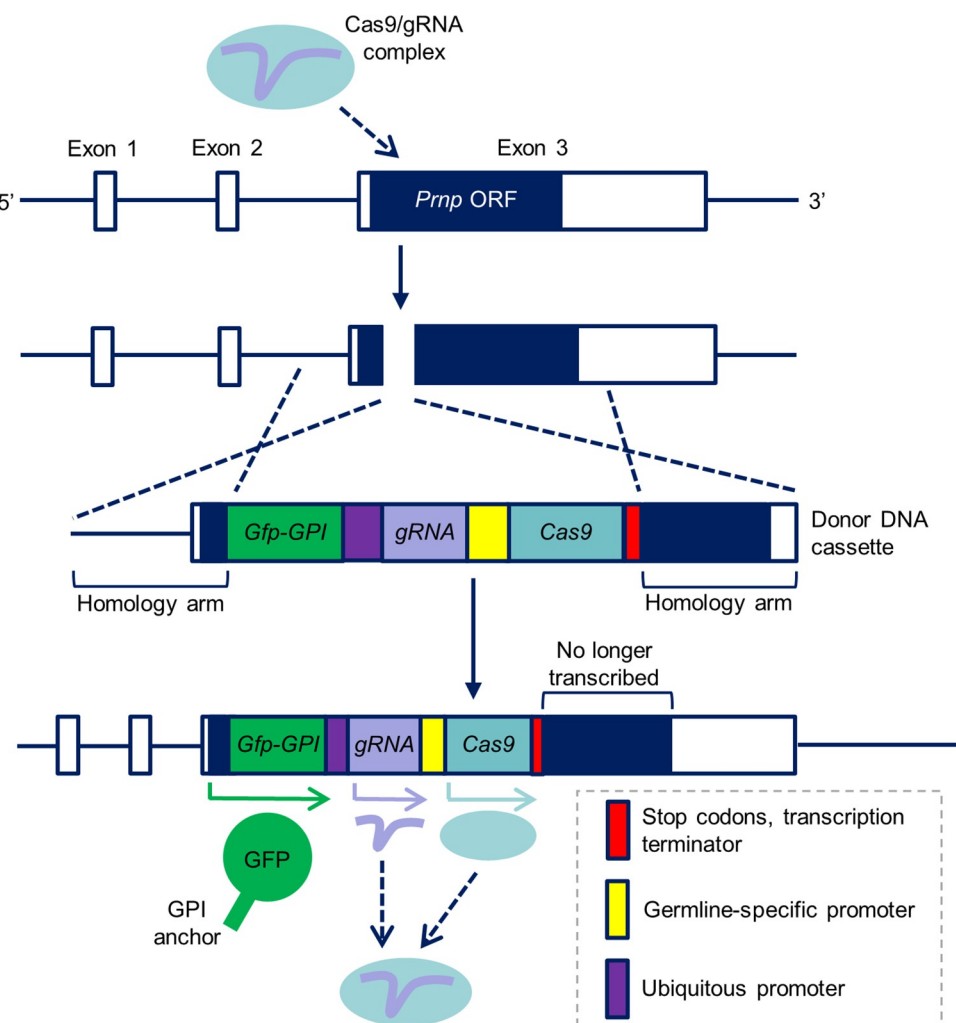

**Fig 1. A CRISPR/Cas9-based gene drive mechanism to spread prion disease resistance.** Diagram showing how Cas9/gRNA ribonucleoprotein complexes would induce a DSB within the *Prnp* ORF that is repaired by HDR in the presence of a donor DNA cassette, leading to the generation of a mobile null *Prnp* allele capable of gene drive. New abbreviation: GPI, glycophosphatidylinositol.

control of germline-specific and ubiquitous promoters (e.g., the U6 RNA polymerase III pro-moter [37]), respectively, as well as a modified GFP reporter transgene (explained in more detail later in the manuscript). *Prnp*-specific homology arms of ~800 bp would flank these sequences; homology arms of this length are recommended when using double-stranded DNA donors [38]. Ribonucleoprotein (RNP) complexes of Cas9 and a *Prnp*-specific gRNA would be delivered to generate a DSB within the *Prnp* ORF, at which point, in the presence of the donor DNA cassette, HDR would result in integration of the donor sequence. This process would create a mobile null *Prnp* allele able to convert the wild type (WT) allele on the sister chromosome in the germline.

A first step was to select suitable gRNA spacer sequences for targeting *Prnp*. The initial codons of the *Prnp* ORF were avoided in case Cas9 cleavage disrupted the splice acceptor site of the protein-coding exon, which can result in chimeric transcripts derived from *Prnp* and extending into the neighbouring gene *Prnd*; this process leads to ectopic expression of Doppel (the protein product of *Prnd*) in the brain, where it is neurotoxic [39]. Therefore, we used a

gRNA design algorithm (no longer available but previously found at http://crispr.mit.edu) to scan codons 23–50 of murine *Prnp*, noting that residues 1–22 comprise the N-terminal signal peptide of PrP[C] that is absent from the mature protein (residues 23–231 in mice). The top scoring sequences were used to generate three slightly different gRNA spacers for testing (S1 Fig, panels A and B). The chosen spacers are a few codons downstream of the sequence used by Mehrabian et al. (2014) to knock out PrP[C] expression in cultured cells [40].

### Selected *Prnp* gRNAs induce Cas9-mediated cleavage within the *Prnp* ORF

Having selected *Prnp*-specific spacer sequences, we prepared RNP complexes consisting of recombinant Cas9 (recCas9) and tracrRNA/crRNA gRNA duplexes. After confirming that recCas9/*Prnp* gRNA–1 RNP complexes induced cleavage of a *Prnp* expression plasmid *in vitro* (Fig 2A), we cloned the spacer sequences separately into the eSpCas9(1.1) vector [41], which is capable of expressing an enhanced-specificity Cas9 (i.e., eCas9) and a single gRNA under control of the CMV and U6 promoters, respectively. Next, we transiently transfected the modified eSpCas9(1.1) vectors into RK13 rabbit kidney epithelial cells stably expressing WT murine PrP[C] (clone WT-5); expression of the endogenous rabbit PrP[C] is undetectable in the RK13 cell line [42]. Forty-eight hours later, we used fluorescence-activated cell sorting (FACS) in an attempt to identify cells with reduced PrP[C] expression due to non-homologous end-joining (NHEJ)-mediated disruptions of the murine *Prnp* ORF. However, expression of *Prnp*-directed gRNAs did not result in obvious reductions in PrP[C] signal intensities when compared with the empty vector control condition. We therefore selected the cells in approximately the lowest decile of PrP[C] signals for each sample (S2 Fig) and performed T7E1 mismatch cleavage assays on the pooled DNA. Faint bands indicative of *Prnp* ORF disruptions were detected for all three gRNAs (Fig 2B), demonstrating that NHEJ events induced by Cas9 cleavage activity had occurred, albeit most likely at low frequency. Equivalent experiments were attempted in the

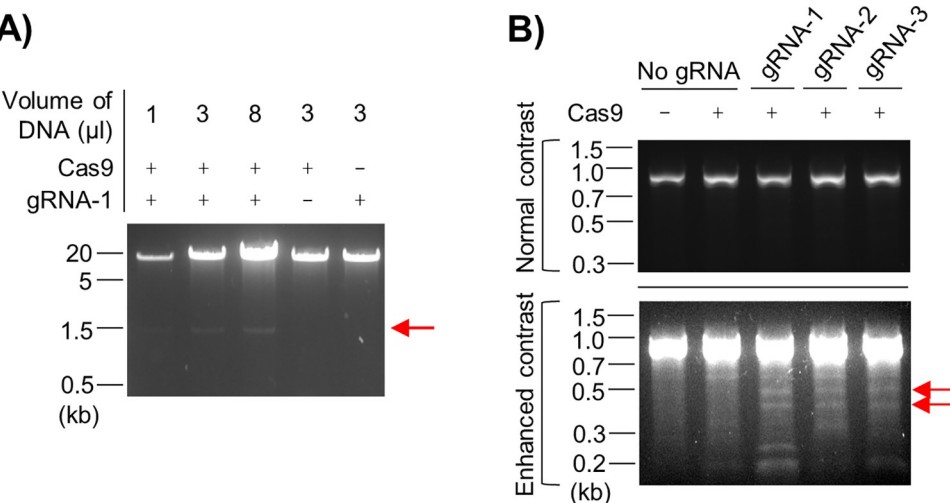

**Fig 2. Selected *Prnp* gRNAs induce Cas9-mediated cleavage within the *Prnp* ORF. (A)** Agarose gel image showing that recCas9 cleaves a half-genomic *Prnp* expression construct (MoPrP.*Xho*.wt) when *Prnp* gRNA–1 is present. A cleavage product of the expected size is indicated by the red arrow. **(B)** Agarose gel images showing the products of T7E1 mismatch cleavage assay reactions for DNA samples obtained from untransfected WT-5 RK13 cells (Cas9-negative) or cells transfected with eSpCas9(1.1) expression plasmids containing *Prnp* gRNA–1, –2 or –3, or no gRNA. In the absence of *Prnp* ORF disruptions, a PCR product of ~900 bp was expected. The faint bands appearing at ~400 and 500 bp only when *Prnp* gRNAs were expressed indicate that indels resulting from NHEJ events were present.

MDB (mule deer brain) cell line [43] with gRNAs directed against the endogenous cervid *Prnp*, but low transfection efficiencies prevented further progress in this regard.

## Integration of a DNA cassette into the *Prnp* locus of murine neuroblastoma cells by HDR of Cas9-induced DSBs

Thus far, we have shown that the selected *Prnp* gRNAs direct Cas9 to cleave within the *Prnp* ORF both *in vitro* and in cell culture. The next step towards the goal of replacing endogenous *Prnp* with a mobile null allele capable of gene drive was to prepare a donor plasmid vector containing a simplified version of the DNA cassette shown in Fig 1, specifically the modified GFP reporter gene flanked by ~800 bp *Prnp* homology arms (Fig 3A). The reporter gene consisted of the GFP coding sequence (minus its start codon) fused to codons 230–254 of murine *Prnp*, which encode the PrP^C C-terminal signal peptide, thereby generating a transgene that we termed *Gfp-GPI*. HDR-mediated integration of the donor DNA cassette should result in expression of GFP driven by the *Prnp* regulatory elements, with the N- and C-terminal signal peptides of PrP^C attached, enabling transit into the secretory pathway and expression at the cell surface via a glycophosphatidylinositol (GPI) anchor attachment, as has been demonstrated previously [44].

N2a murine neuroblastoma cells were co-transfected with the donor vector and eSpCas9 (1.1) expression vectors containing *Prnp* gRNAs. Unfortunately, we were unable to detect cells expressing the reporter using fluorescence microscopy to examine bulk cell populations. Nonetheless, integration of the *Gfp-GPI* transgene did occur, at least in some cells, because 3' junction PCR analysis of pooled DNA samples collected 3 days after transfection resulted in

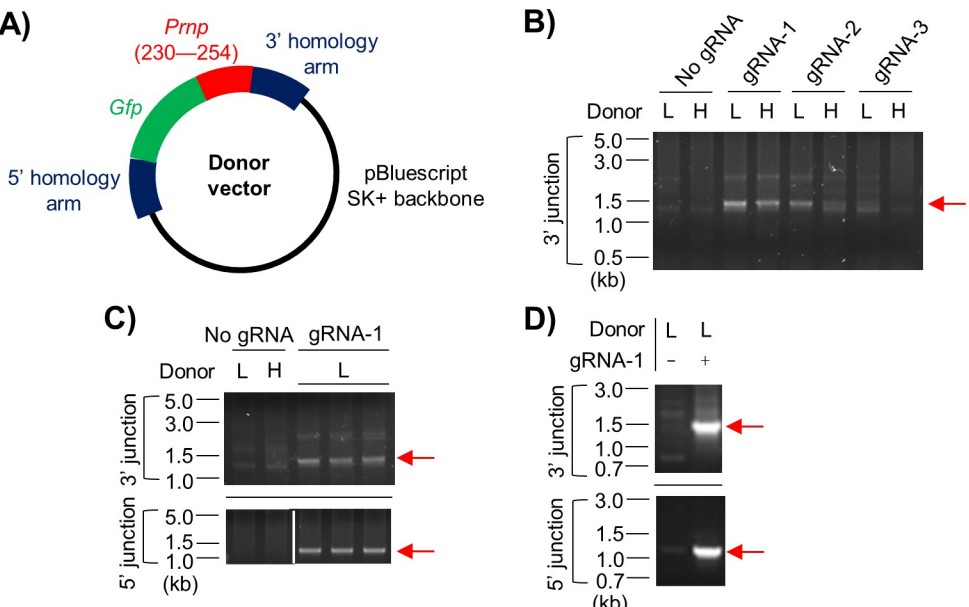

**Fig 3. Integration of a DNA cassette into the *Prnp* locus of murine neuroblastoma cells by HDR of Cas9-induced DSBs. (A)** Diagram of the donor vector generated for HDR experiments. *Prnp* homology arms flank a promoter-less reporter gene consisting of the *Gfp* ORF fused to the 3' end of the *Prnp* ORF that would normally encode residues 230–254 of PrP^C. **(B–D)** Images from agarose gel electrophoresis of junction PCR products showing that co-transfection of N2a cells with the donor vector and eSpCas9(1.1) expression plasmids containing *Prnp* gRNAs results in integration of the *Gfp-GPI* transgene into the *Prnp* locus, at least in some cells. DNA was isolated 3 days **(B, C)** or 6 days **(D)** after transfection. "L" and "H" denote low (1:1) and high (5:1) ratios of donor vector to eSpCas9(1.1), respectively. Red arrows indicate the presence of PCR products of the expected sizes. The white line on the bottom part of panel C indicates that irrelevant lanes have been omitted from the image.

bands of the expected size only when *Prnp* gRNAs were present ([Fig 3B]). A follow-up 5' junction PCR analysis for *Prnp* gRNA–1, the best performing of the three gRNAs in the initial experiment, also resulted in bands indicative of donor DNA integration ([Fig 3C]). Providing more time after transfection for editing to occur (6 days) appeared to result in more intense bands in the diagnostic PCR for the 5' and 3 junction fragments ([Fig 3D]), although a side-by-side numerical comparison of the two conditions was not performed.

Even in the presence of a donor DNA cassette, Cas9-induced DSBs may still be repaired by NHEJ on some occasions. Indels derived from these NHEJ events could make the *Prnp* alleles permanently resistant to the Cas9/gRNA combination. Alternatively, the presence of small indels may still permit subsequent HDR events to occur; depending on the nature of the repair, the reporter transgene may not function effectively or the CRISPR protospacer sequence within the endogenous *Prnp* may not be split as intended, potentially rendering the null *Prnp* allele itself vulnerable to new rounds of Cas9-mediated cleavage. Because of these issues we proceeded to check whether integration of the donor DNA cassette had occurred precisely in the N2a cells. The junction PCR products shown in [Fig 3C] were gel-purified, cloned into pCr2.1-TOPO, and transformed into *E. coli*. Twelve clones in total were selected, six for each junction. Diagnostic restriction digests confirmed the presence of an insert of the correct size in 5/6 cases for the 3' junction and 6/6 for the 5' junction ([S3 Fig], panel A). Sanger sequencing of four of the 5' junction clones and three of the 3' junction clones with normal restriction digest profiles showed that the expected sequence was present with no indels detected around the junction sites ([S3 Fig], panels B and C). The 3' junction clone with the abnormal restriction digest profile (#3) was also sequenced, but the data were hard to interpret, with tracts of the expected sequence identified together with other unknown sequences. Nonetheless, considered as a whole, these data show that we succeeded in integrating a ~1.2 kb DNA sequence (the *Gfp-GPI* transgene) into the *Prnp* ORF of N2a cells and that this integration occurred precisely in the majority of cases.

## Generation of CRISPR/Cas9-induced disruptions of the *Prnp* ORF in mice

Following the successful demonstration of *Prnp* editing in cell culture, we tested the effectiveness of our chosen *Prnp* gRNAs *in vivo*. The results of these experiments are summarized in [Table 1]. Two attempts at microinjecting recCas9/*Prnp* gRNA–3 RNP complexes (tracrRNA/crRNA gRNA duplex format) into fertilized FVB/NJ mouse oocytes were unsuccessful; sequencing of 31 viable pups revealed no signs of *Prnp* ORF disruptions. However, electroporation of RNP complexes, yielding a combined 18 live births from two experiments, led to the

**Table 1. Summary of gene editing events.**

| Experimental configuration | Allele yield[a] | Live mice obtained | Allele type | Inferred mechanism |
|---|---|---|---|---|
| Microinjection of oocytes with RNP complexes | 0/31 (0/62) | 31 | N/A | N/A |
| Electroporation of oocytes with RNP complexes | 3/18 (3/36) | 18 | 1 bp deletion (frameshift) | Classical NHEJ |
| | | | 6 bp deletion (in-frame) | Alternative NHEJ (MMEJ) |
| | | | 21 bp deletion (in-frame) | Alternative NHEJ (MMEJ) |
| Electroporation of oocytes with RNP complexes | 1/9 (1/18) | N/A (embryos harvested at 5.5 dpc) | 48 + 27 bp deletions (both in-frame) | 2 alternative NHEJ (MMEJ) events |

See [Fig 4] and [S4 Fig] for more detailed results from these experiments. *Prnp* gRNA–3 was used in each case. New abbreviations: MMEJ, microhomology-mediated end-joining; dpc, days post coitus.

[a]The first value represents the modification frequency (i.e., successful modifications over the number of mice or mouse embryos screened). Numbers in parentheses are adjusted for the number of allele targets per diploid genome.

identification of three founders with disruptions to one *Prnp* allele, as determined by Sanger sequencing (Fig 4A). Two founders (#34 and #36) had in-frame deletions (21 bp and 6 bp) in the vicinity of the expected Cas9 cleavage site; the presence of a disrupted *Prnp* allele in founder 34 was confirmed by T7E1 assay (Fig 4B). These in-frame deletions remove amino acids lying within the first of PrP$^C$'s two hexarepeats of the form GGN/SRYP. An additional founder (#33) had a frameshift mutation resulting from an insertion of 1 bp that was predicted to generate an abnormal sequence from codon 39 onwards with a new stop codon at position 78. Mice derived from each of the founders appeared phenotypically normal, although detailed characterization was not performed. Homozygous progeny were obtained for line 33 and a lack of PrP$^C$ expression in the brain was confirmed by capillary western analysis (Fig 4C). Furthermore, we obtained Sanger sequencing data from PCR amplicons containing three of the highest-scoring potential off-target sites for gRNA–3 that were identified by the gRNA design algorithm (S1 Fig, panel C), but no indels were detected for any of the three founders.

Because the deletion in founder 34 was quite large (21 bp), we wanted to determine whether such deletions would be commonplace. To simplify the process, Cas9/*Prnp* gRNA–3 RNP complexes were electroporated into fertilized oocytes as before, but this time the embryos were grown *in vitro* to 5.5 days post coitus (dpc) before being used for preparation of genomic DNA and subsequent direct sequencing of PCR products. Although sequencing data of suitable quality were obtained for nine embryos, only one had a disrupted *Prnp* allele. However, this allele had an unusual double deletion of 48 and 27 bp, with a short tract of *Prnp* sequence retained in between (S4 Fig).

## Transgenic mice expressing *Cas9* under control of the *Prl3b1* promoter do not express Cas9 in the germline

Having created mice with Cas9-induced *Prnp* disruptions, we attempted to engineer mice capable of expressing Cas9 in the germline–an important step towards creating a fully mobile genetic element for gene drive. We searched for a male germline-specific promoter, because findings from an earlier study in mosquitos suggested that expression in the female germline can lead to Cas9 persistence in the egg, causing NHEJ-induced indels that render the target gene resistant to the drive mechanism [32]. The gene *Prl3b1* was chosen due to its reported activity in the germline of male mice [45], and transgenic (Tg) mice expressing eCas9 under control of the *Prl3b1* promoter (*Prl3b1-eCas9*$^{+/-}$) were generated. However, we were unable to detect expression of eCas9 in various male urogenital tissues of *Prl3b1-eCas9*$^{+/-}$Tg mice derived from founder 4 (Fig 5A). Testes isolated from mice derived from other founders were also analysed, as were female reproductive tract tissues (ovaries) from the same mice, but no Cas9 expression was observed (Fig 5B). In addition to its reported activity in the male germline, *Prl3b1* is expressed in the placenta [46]. We therefore crossed *Prl3b1-eCas9*$^{+/-}$and WT mice and prepared homogenates of embryos extracted from 15 dpc pregnant females together with their respective placentas. Cas9 expression was detected in approximately half of the embryo-placenta homogenates (Fig 5C), which indicates that the transgene construct was functional and that the problem clearly lay with the promoter, which was not sufficiently active in the male germline for our purposes.

## Discussion

### Steps towards population-based eradication of prion disease

In this study, we embarked on a series of steps to develop a complete CRISPR/Cas9-based gene drive mechanism able to promote the spread of a null *Prnp* allele among a population of mice and hence confer resistance to prion infection. While this ultimate goal was not achieved,

**A)**

CACCGGTGGAAGCCGGTATCCCGGGCAGGGAAGCCCTGGAGGC
140        150        160        170

WT mouse *Prnp*

Expected Cas9 cleavage site

CACCGGTGGAAGCCGG**TATNCCNGGNNNGGAAANCCNGGNNGN**
140        150        160        170

Line 33

1 bp insertion in one *Prnp* allele

CACCGGTGGAAGCC**GG**NA**TCCCGGGCAGG**N**AANC**CCTG**N**ANGN**
140        150        160        170

Line 34

Start of 21 bp deletion in one *Prnp* allele

CACCGGTGGAA**NCCNNNNTCCCGGNANGNNNNNNCCTG**N**ANGN**
140        150        160        170

Line 36

Start of 6 bp deletion in one *Prnp* allele

**Translating to protein**

```
              33                    50
              |                     |
    WT        TGGSRYPQGGSPGGNRYP
                                    78
                                    |
    Line 33   TGGSRY...//...stop
                                    50
                                    |
    Line 34   TGGSRYPGQGSPGGNRYP
    Line 36   TGGSRYPGQGSPGGNRYP
```

**B)**

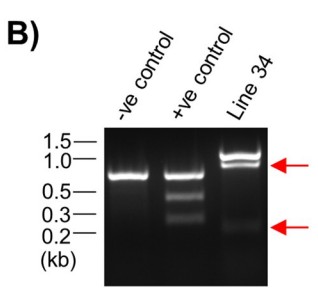

**C)**

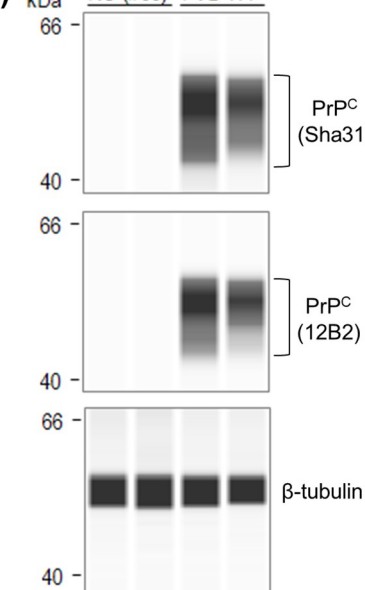

**Fig 4. Generation of CRISPR/Cas9-induced disruptions of the *Prnp* ORF in mice. (A)** Sanger sequencing chromatograms for mice derived from fertilized FVB/NJ oocytes that were electroporated with recCas9/*Prnp* gRNA–3 RNP complexes. The numbers below the automated base calls indicate the position in the sequencing read rather than the *Prnp* ORF. Changes to the PrP^C amino acid sequence for the lines with disrupted *Prnp* are indicated in the bottom box. **(B)** Agarose gel image showing the products of T7E1 mismatch cleavage assay reactions for negative (homoduplex) and positive (heteroduplex) control DNA solutions and DNA obtained from the founder of line 34. The presence of the bands indicated by the red arrows confirms that line 34 has a disrupted *Prnp* allele. **(C)** Capillary western images confirming that PrP^C expression is undetectable in brain homogenates from homozygous line 33 mice. Sha31 and 12B2 are two different PrP^C antibodies. New abbreviation: KO, knockout.

we nonetheless showed that our selected *Prnp* gRNAs were active in mice and that we could modify the *Prnp* locus in N2a cells by introducing a donor DNA sequence via HDR of Cas9-induced DSBs. The reagents generated for the cell experiments could be repurposed to perform different edits to the *Prnp* locus, such as the introduction of disease-relevant mutations in order to study their effects. This approach would enable levels of the mutant *Prnp* to be controlled by the endogenous regulators of *Prnp* expression, in contrast to the more basic strategy of knocking out *Prnp* using CRISPR/Cas9 followed by random integration into the genome of a transgene construct or lentiviral vector containing the mutant *Prnp*.

The T7E1 assay data shown in Fig 2B suggest that a relatively low proportion of the RK13 cells contained disruptions to the murine *Prnp* ORF. In these experiments, the cells were harvested two days after transfection with the Cas9/gRNA expression vectors. As we discovered through subsequent experiments performed in N2a cells, harvesting the cells after a longer post-transfection time interval may have improved the cleavage efficiency. In addition, although improving gene editing performance in cultured cells was not the main focus of this study, co-treatment with cell cycle-regulating compounds would likely lead to further gains in Cas9 cleavage efficiency [47]. Similar experiments performed in MDB cells using gRNAs targeting the endogenous cervid *Prnp* were unsuccessful due to low transfection efficiencies; lentiviral transduction of CRISPR/Cas9 components might be a more effective approach for this cell line. Alternatively, experiments could be performed using an easily transfectable cell line (e.g., HEK293T) engineered to contain cervid *Prnp*.

Returning to the N2a cell experiments, integration of the donor DNA cassette into the genome typically occurred in a precise manner, with no errors detected around the junction sites in most cases. Although we expected precise integration of the donor cassette to result in expression of a GPI-anchored GFP reporter together with elimination of PrP^C expression, we did not detect any green-fluorescent cells, perhaps because endogenous PrP^C expression is relatively low in the N2a cell line (and expression of the reporter would be under control of the *Prnp* promoter) [48]. We did not assess whether the production of PrP^C from edited alleles was eliminated, because N2a cells are reportedly at least tetraploid [49], on average, so disruption to one *Prnp* allele would be unlikely to reduce PrP^C expression sufficiently for overt differences to be detected by immunofluorescence microscopy.

During testing of the selected *Prnp* gRNAs *in vivo*, we obtained a founder (#33) with a 1 bp insertion within the *Prnp* ORF resulting in a frameshift from codon 39 onwards, with a premature stop codon created at position 78. Although PrP^C expression was not detected in homozygous progeny of founder 33 using the antibodies Sha31 (epitope 144–151; [50]) and 12B2 (epitope 88–92; [51]), further characterization is required to determine if the truncated protein encoded by the mutant *Prnp* allele is actually expressed and, if so, whether there are any gain-of-function effects. We would, however, expect line 33 to exhibit complete loss of normal PrP^C function, given that only 16 of the remaining 38 unmodified codons are actually present in the mature protein (due to removal of the N-terminal signal peptide). We believe that these mice may be useful for other laboratories studying prion disorders, noting that a similar coisogenic

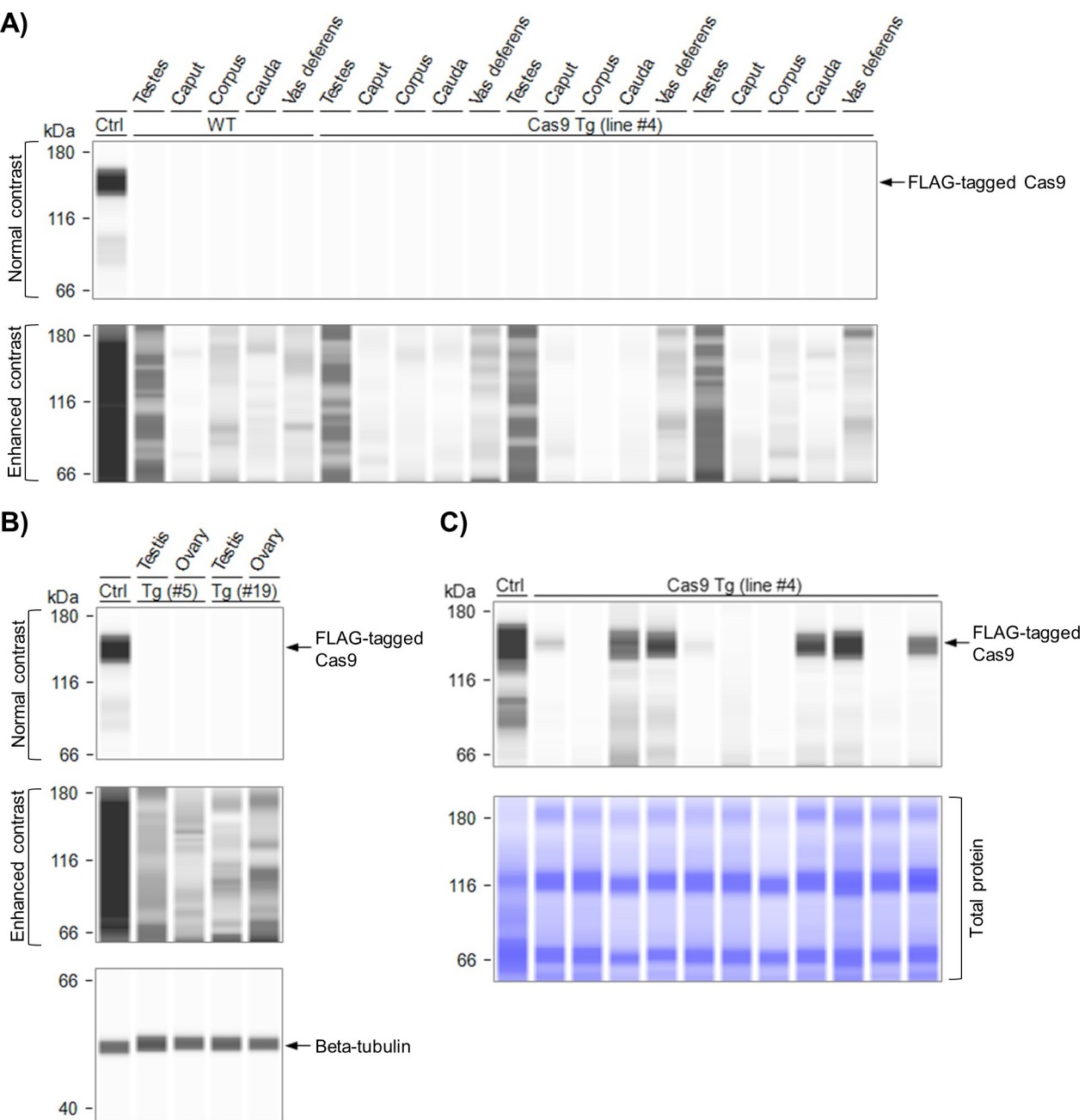

**Fig 5. Transgenic mice expressing *Cas9* under control of the *Prl3b1* promoter do not express Cas9 in the germline. (A–C)** Capillary western images showing that Cas9 expression was not detected in male urogenital tract tissues of *Prl3b1-eCas9*$^{+/-}$ Tg mice (n = 3) derived from founder 4 (**A**) or in testes and ovaries of mice derived from founders 5 and 19 (**B**) but was detectable in 15 dpc embryo-placenta homogenates of line 4 (**C**). The positive control ("ctrl") was a lysate of RK13 cells transiently expressing Cas9. Caput, corpus and cauda are sections of the epididymis. The embryo-placenta tissues (n = 11) were obtained from two litters resulting from crosses between *Prl3b1-eCas9*$^{+/-}$ and WT mice (therefore, ~ half of the tissues would be expected to be positive for Cas9).

knockout line (*Prnp*$^{ZH3/ZH3}$) was recently generated using a transcription activator-like effector nuclease [29], an alternative gene editing methodology. In line with prior studies, we have designated the FVB/NJ mice bearing this frameshifted allele as *Prnp*$^{Edm/Edm}$.

With regards to the goal of functional expression of eCas9 in the germline, we found that use of the *Prl3b1* promoter in a transgene construct did not yield expression in the germline of

male mice in contrast to a previous report [45]. We had searched for male germline-specific promoters due to earlier findings that Cas9 expression in the female germline of *Anopheles stephensi* mosquitos may lead to Cas9 persistence in the egg, causing NHEJ-induced indels that render the target gene resistant to the drive mechanism [32]. However, after we had begun our mouse transgenesis experiments, the first demonstration of gene drive in a mammalian species (the mouse) was published by Grunwald et al. (2019) [52]. Interestingly, the authors of this study found that the driving allele was only able to propagate when Cas9 was expressed in the female germline–the opposite result to the mosquito study. Moreover, even when crosses with the male Cas9-expressing mice are excluded from the calculation, the most effective genetic strategy tested achieved a drive efficiency of only 44%, which is likely insufficient for spreading null alleles in a wild population, particularly given that mammals have much longer generation times than insects. The authors speculated that the low efficiency may have been because precise matching of the timing of Cas9 expression to the window in which meiotic recombination occurs is more important in mammals than in insect species. Given these findings and the lack of additional reports of mammalian gene drive since the study by Grunwald et al. (2019) was published, the practicality of this approach in mammals remains unclear. Nevertheless, in the absence of suitably effective control strategies for CWD, we consider it important to explore all possible solutions, including gene drive.

## CRISPR-induced alleles and genetic resistance to prion infection

Further work on the aforementioned genetic strategies raises the question of the optimal *Prnp* allele to pursue for *in vivo* experiments and the fine-structure repertoire of CRISPR-induced mutations. Surprisingly, we found a preponderance of in-frame deletions in our sample set that would affect the N-terminal natively disordered region of PrP$^C$ but would spare the globular C-terminal domain. In addition to the 1 bp insertion identified in founder 33, we detected three *Prnp* alleles with in-frame deletions of varying size when testing the *Prnp*-directed gRNAs *in vivo*. Repair of WT Cas9-induced DSBs by classical NHEJ is either error-free or responsible for very small indels, with a bias towards deletions and a median size of 3 bp reported previously [53]. However, two of the mutated alleles we identified contained rather large deletions: 21 bp in one allele and a double deletion of 48 bp and 27 bp in the other. Such deletions are more likely to be caused by an alternative NHEJ pathway called microhomology-mediated end-joining, which can repair DNA when end resection leads to single-strand overhangs with microhomologies of $\geq$ 5 bp [54, 55]; for each of the larger deletions we obtained, perfect microhomologies of 6–8 bp are present immediately 5' of the deleted sequence and at the 3' end of the deleted sequence. Nonetheless, the mechanism behind the double deletion is not clear, because the 27 bp deleted sequence starts almost 50 bp away from the predicted Cas9 cleavage site.

Microhomologies occur frequently in the sequence encoding the N-terminal domain of PrP$^C$ due to the presence of amino acid repeats in this part of the protein (GGN/SRYP hexarepeats and PHGGG/SWGQ octarepeats), with the codon usage often being similar. Therefore, the microhomologies are usually in frame with each other, which would explain why the large deletions we obtained were, in turn, in frame. This effect is problematic for a gene drive mechanism based on the spread of a null *Prnp* allele, because DSBs will sometimes be repaired by mechanisms other than HDR even in the case of a highly efficient drive; in such situations, classical NHEJ repair would be ideal, because it is likely to produce frameshift mutations that would eliminate the production of PrP$^C$. In contrast, the large in-frame deletions we have observed would generate permanent resistance to the drive mechanism (due to disruption of the CRISPR protospacer sequence) without greatly affecting the protein product of the gene

and, by extension, susceptibility to CWD. Certain PrP deletion mutants (e.g., Δ32–121, Δ32–134, Δ94–134) are associated with lethal neurodegenerative phenotypes in mice [56, 57], but these phenotypes require the deletion to extend into the hydrophobic core of the protein (residues ~112–133). None of the deletions generated in our study fit this criteria, although we cannot completely exclude the possibility that sufficiently large deletions could be created. To avoid generating these in-frame deletions, the *Prnp* gRNA could be targeted to a more 3' prime position away from the microhomology-dense region, but this would risk generating toxic truncated protein products similar to the pathogenic stop codon allele Y145X [58]. As an alternative to null *Prnp* alleles entirely, protective amino acid substitutions could be introduced by deploying two gRNAs simultaneously to target sequences 5' and 3' of the substitution site in order to produce a modified coding sequence by a double homologous recombination approach [59]–akin to the method used to make knock-in mice. Variant *Prnp* ORFs to be considered might include the G127V polymorphism (equivalent to G130V in cervid PrP$^C$), which appears to be protective against all human prion strains [60]. Avoiding fully penetrant null alleles would also alleviate the potential for loss of PrP$^C$ expression to cause the peripheral neuropathy phenotype that has been identified in PrP$^C$-null mice and goats [28–30]. Gene drive-enabled variant *Prnp* alleles could then be mobilized in tandem with "gene brake" safety measures [61, 62], which are likely to be necessary for safe use of gene drive a real world setting and could be important for gaining general public acceptance [63].

In conclusion, although we encountered various technical challenges, including issues with obtaining germline expression of Cas9, that prevented us from developing a complete gene drive mechanism capable of spreading resistance to prion infection, we did succeed in generating reagents able to modify the murine *Prnp* locus in cultured cells via HDR of Cas9-induced DSBs and created a line of PrP$^C$-knockout mice that could be useful to other researchers in the field. Our study has also helped to underline the challenges associated with gene drive in mammalian species, allowing us to pivot towards investigating other potential approaches for controlling CWD and prion disorders in general.

## Materials and methods

### Selection of suitable spacer sequences for Prnp gRNAs

Codons 23–50 of murine *Prnp* were scanned for suitable gRNA spacer sequences using an algorithm developed in the laboratory of Dr. Feng Zhang (no longer available but previously found at http://crispr.mit.edu).

### In vitro cleavage of MoPrP.Xho.wt construct

RecCas9 (Integrated DNA Technologies [IDT], 147821858), CRISPR-Cas9 tracrRNA (IDT, 147821857) and a CRISPR-Cas9 crRNA containing *Prnp*-specific spacer sequence #1 (IDT; see S1 Fig, panel B, for sequence) were combined at equimolar amounts, heated to 95˚C for 5 min and cooled to room temperature to create RNP complexes. A linearized "half-genomic" *Prnp* expression construct (MoPrP.*Xho*.wt) [42] was incubated with Cas9/gRNA RNP complexes (100 nM) for 1 h at 37˚C in a reaction buffer of 20 mM HEPES, pH 6.5, 100 mM NaCl, 5 mM MgCl$_2$, and 0.1 mM EDTA. Reactions were stopped by incubation with Proteinase K (Roche, 3115887001) at ~1 mg/mL for 10 min at 56˚C.

### Agarose gel electrophoresis

DNA samples were separated by agarose gel electrophoresis with the GeneRuler 1 kb Plus DNA Ladder (Thermo Scientific, SM1331) run alongside. Gels were stained with

SYBR Safe (Invitrogen, S33102) and images obtained using a Fluor Chem E imager (ProteinSimple).

## Molecular cloning

The eSpCas9(1.1) expression vector was a gift from Dr. Feng Zhang (Addgene plasmid #71814; http://n2t.net/addgene:71814; RRID:Addgene_71814) [41]. Sense and antisense oligo-nucleotides matching the selected gRNA spacer sequences (S1 Fig, panel B) were synthesized by IDT, annealed and cloned separately into eSpCas9(1.1) using *Bbs*I.

Generation of the donor vector for HDR consisted of several stages; sequences of the relevant primers and DNA fragments are provided in S1 Table. In stage 1, a gBlock fragment (IDT) containing codons 230–254 of murine *Prnp* was joined to a PCR product derived from pBud.GFP using a Gibson Assembly Cloning Kit (New England Biolabs, E5510S), thereby inserting the *Prnp* fragment just before the *Gfp* stop codon and generating the *Gfp-GPI* transgene. In stage 2, a ~1.6 kb fragment consisting of the *Prnp* ORF flanked by 5' (intron 2) and 3' sequences (3' untranslated region) was amplified from WT FVB/NJ mouse genomic DNA and cloned into pBluescript SK+ using *Sal*I and *Hind*III (to make pBluescript SK+.PrnpHA). The primers for this step were designed so that the protospacer and protospacer-adjacent motif for *Prnp* gRNA–3 were added at both ends of the *Prnp* fragment, thereby promoting excision of the DNA cassette in the presence of Cas9 and the *Prnp* gRNA; this approach has been reported to improve HDR efficiency significantly [47]. In the third and final stage, a PCR product consisting of the *Gfp-GPI* transgene (minus the CMV promoter and start codon but including the 3' bovine growth hormone polyadenylation signal sequence from the pBud vector) was joined using Gibson Assembly to a PCR product consisting of the entire pBluescript SK+.PrnpHA vector in linear form. Thus, the *Gfp-GPI* transgene was inserted approximately into the middle of the 1.6 kb *Prnp* fragment, thereby splitting the CRISPR protospacer sequence and generating ~0.8 kb homology arms on either side to promote integration by HDR.

Generation of the construct used to make *Prl3b1-Cas9* transgenic mice also consisted of several stages; sequences of the relevant primers are provided in S1 Table. In stage 1, the bovine growth hormone polyadenylation signal sequence was amplified from pBud.CE4 (Invitrogen) and cloned into pBluescript SK+ using *Xma*I and *Spe*I. In stage 2, in order to add an intron upstream of the site the Cas9 cDNA was to be inserted, we used rabbit genomic DNA to amplify a region of the β-globin gene and cloned it into the construct from stage 1 using *Hind*III and *Pst*I. In stage 3, an insulator sequence was added to reduce potential positional silencing. We used the human β-globin locus control region, amplified from BAC CTD 3055E11, and cloned it into the construct from stage 2 using *Kpn*I and *Xho*I. The forward primer used for this step contained a *Not*I site downstream of the *Kpn*I site to aid in purifying the final transgene from the vector backbone. In stage 4, to help with later cloning steps, we added a linker sequence with phosphorylated overlapping oligonucleotides between *Xho*I and *Cla*I sites. This step added *Asc*I, *Nde*I, *Aat*II and *Nhe*I recognition sequences. In stage 5, we added an *Age*I site to the vector using phosphorylated overlapping oligonucleotides between *Pst*I and *Xma*I sites. In stage 6, the Cas9 coding region was excised from eSpCas9(1.1) using *Age*I and *Not*I and cloned into the construct from stage 5 using the same restriction enzymes. In the seventh and final stage, the *Prl3b1* promoter region was amplified by nested PCR from mouse BAC RP23-189A16 and cloned into the *Xho*I and *Nhe*I sites of the construct from stage 6.

All plasmids were purified at the final stage using an EndoFree Plasmid Maxi Kit (Qiagen, 12362). Sequences were checked by diagnostic restriction digests and, subsequently, by Sanger sequencing. The donor vector for HDR was found to contain a 7 bp deletion (of a repetitive tract of G nucleotides) within the 5' *Prnp* homology arm. However, this error was >200 bp

upstream of the expected Cas9 cleavage site and so was considered unlikely to affect HDR efficiency to any great extent.

## Cell culture and transfections

RK13 cells stably expressing WT murine PrP$^C$ (clone WT-5) had been generated from the parental RK13 cell line (ATCC, CCL-37) as part of an earlier study [42]. WT-5 RK13 and N2a cells (ATCC, CCL-131) were routinely cultured at 37˚C in 5% CO$_2$ with 95% humidity in DMEM (Gibco, LS11885084) containing 10% (v/v) fetal bovine serum (FBS; Life Technologies, LS12483020) and 1% (v/v) penicillin-streptomycin solution (pen-strep; Gibco, LS15140122). Transient transfections were performed using Lipofectamine 2000 (Invitrogen, 116608027) or Lipofectamine 3000 (Invitrogen, L3000008) 24 h after cells were seeded into plates in DMEM supplemented with 10% (v/v) FBS.

## Fluorescence-activated cell sorting and DNA isolation

Cells were detached using Accutase (Corning, 25058C) and transferred to collection tubes (Corning, 352058). Working on ice, cells were washed, blocked (10 min), incubated with Zombie Aqua dye (Biolegend, 423101; diluted 1:250 from the reconstituted DMSO stock), incubated with SAF83 anti-PrP antibody (Cayman Chemicals, 189765; diluted 1:50; 20 min), washed again, incubated with goat anti-mouse Dylight 649 secondary antibody (Thermo Scientific, 35515; diluted 1:100; 20 min), and washed once more. PBS (Boston Bioproducts, BM-220) supplemented with 25 mM HEPES, 1% (v/v) FBS, 0.5 mM EDTA, and 1% (v/v) pen-strep was used as the buffer for all the above steps (except for adjustment of FBS to 10% for the blocking step). Solutions were exchanged by centrifugation followed by resuspension of the pellet in the new buffer. FACS based on PrP$^C$ signal intensity was performed by staff at the University of Alberta's Flow Cytometry Facility using a BD FACSAria III instrument, with FACSDiva Version 6.1.3 (BD Biosciences) used for analysis. The intense fluorescence of GFP produced by the stably integrated expression vector interfered with Zombie Aqua-based live/dead cell discrimination. Instead, tight gating based on side scatter (SSC) area vs forward scatter (FSC) area plots was used to exclude dead cells. Additional gates based on SSC and FSC width vs height plots were applied to exclude cell clumps. A final gate was applied to select the ~10% of cells with the lowest PrP$^C$ signals, which were centrifuged, resuspended in PBS containing 25 mM HEPES, and centrifuged again. DNA was isolated from cell pellets using a DNAeasy Blood and Tissue Kit (Qiagen, 69504).

## T7E1 mismatch cleavage assays

*Prnp* fragments were amplified using AccuPrime Taq DNA Polymerase, High Fidelity (Invitrogen, LS12346086; see S1 Table for primer sequences). After agarose gel electrophoresis, DNA concentrations were estimated by comparison to the GeneRuler 1 kb Plus DNA Ladder; band intensities were quantified using ImageJ. Hybridization and T7E1 incubation steps were performed according to the Alt-R Genome Editing Detection Kit instructions (Integrated DNA Technologies, 1075931), except that incubation with T7E1 was shortened to 30 min. Reactions were stopped by addition of EDTA to ~25 mM.

## Junction PCRs

PCRs were performed using AccuPrime Taq DNA Polymerase, High Fidelity (Invitrogen, LS12346086) and primers (see S1 Table) that would only generate an amplicon if the *Gfp-GPI* transgene insert was present. PCR products were extracted from agarose gels using a MinElute

Gel Purification Kit (Qiagen, 28604), ligated into the pCR2.1–TOPO TA vector using a cloning kit (Invitrogen, 450641) and transformed into One Shot Max Efficiency DH5α-T1$^R$ Competent Cells (Invitrogen, 12297016). Plasmid DNA samples were subjected to diagnostic restriction digests followed by Sanger sequencing.

## Generation of transgenic mice

Animal handling procedures and husbandry were in accordance with Canadian Council on Animal Care guidelines and approved by the University of Alberta institutional ethics review (AUP00000356, AUP00000358). To test *Prnp* gRNAs *in vivo*, RNP complexes were prepared as described previously for the *in vitro* cleavage experiment, except that the crRNA contained *Prnp*-specific spacer sequence #3 (see S1 Fig, panel B, for sequence). RNP complexes were electroporated into fertilized FVB/NJ mouse oocytes at the University of Alberta's Transgenic Core Facility. Zygotes were washed into 10 μL Opti-MEM (Gibco, 31985062) and mixed with the RNP complexes in 10 μL Opti-MEM. The mixture was transferred to a 0.1 cm electroporation cuvette (Bio-Rad, 1652083). Electroporation was performed in a Bio-Rad GenePulser XCell using three pulses of 3.0 ms at 30V with a 100 ms interval between pulses. Zygotes were cultured overnight in KSOM media (Millipore, MR-121-D) in a MINC benchtop incubator (Cook Medical; 37˚ C, 5% $CO_2$, and 5% $O_2$/nitrogen). Viable embryos were surgically transferred to pseudopregnant CD1 female mice on the following day, except for when *Prnp* was sequenced in blastocyst-stage embryos; in these cases, embryos were cultured *in vitro* to 5.5 dpc before being washed with M2 medium (Sigma-Aldrich, M7167) and transferred individually to PCR tubes. Embryos were digested for 30 min at 56˚C in 10 μl of PBDN buffer ("PCR Buffer Nonionic Detergents" [64]), consisting of 50 mM KCl, 10 mM Tris, pH 8.3, 2.5 mM $MgCl_2$, 0.1 mg/ml gelatin, 0.45% (v/v) NP-40 substitute, 0.45% Tween-20, and 0.1 mg/ml proteinase K (Invitrogen, 25530049). After incubating at 95˚C for 10 min to inactivate proteinase K, amplification of *Prnp* by PCR was performed using AccuPrime Taq DNA Polymerase, High Fidelity (Invitrogen, LS12346086; primers listed in S1 Table).

*Prl3b1-Cas9* transgenic mice were generated by microinjection of the purified construct into the pronuclei of 0.5 dpc FVB/NJ fertilized oocytes using a XenoWorks digital microinjector (Sutter). Viable zygotes were surgically transferred to psuedopregnant CD1 female mice.

## Isolation and purification of DNA from mice

Ear or tail tissue samples were digested overnight at 55˚C with 0.8 mg/mL proteinase K (Invitrogen, 25530049) in a buffer of 50 mM Tris, pH 8.0, 100 mM NaCl and 1% (w/v) SDS. An equal volume of Buffer-Saturated Phenol was added (Invitrogen, 15513047) and samples were centrifuged at $16000 \times g$. The DNA was precipitated using 95% (v/v) ethanol, centrifuged, washed with 70% ethanol and resuspended in TE buffer.

## Sanger sequencing

Sanger sequencing of DNA samples was performed by staff at the Molecular Biology Services Unit of the Department of Biological Sciences, University of Alberta (see S1 Table for details of sequencing primers). DNA chromatograms were checked using SnapGene Viewer version 6.0.2 and sequence alignments were performed using Serial Cloner version 2.6.1.

## Homogenization of tissues and capillary western assays

Adult mice were euthanized by cervical dislocation and tissues were immediately extracted and frozen on dry ice. Embryos together with their respective placentas were extracted from

pregnant females at 15 dpc. Mouse brains were homogenized as described previously [65]. Other tissue types were homogenized in a buffer of 50 mM Tris, pH 7.4, 150 mM NaCl, 1% (v/v) NP-40 substitute, 0.5% (w/v) sodium deoxycholate, 0.1% (w/v) SDS, 1 mM EDTA, and a protease inhibitor cocktail (Roche, 04693159001) using needles of decreasing diameter (urogenital tract tissues) or a Dounce homogenizer (embryo–placenta tissues). Homogenates of all tissues other than brain were clarified by centrifugation and total protein concentrations in the supernatants were determined by bicinchoninic acid assay (Thermo Scientific, 0023225). Capillary western plates were prepared, run using a Wes instrument (ProteinSimple) and analysed using the associated Compass software as described previously [65]. Primary antibodies were Sha31 anti-PrP (Spi-Bio Inc., A03213; diluted 1:10000), 12B2 anti-PrP (see Ref. [51]; diluted 1:500) and anti-beta-tubulin (Novus Biologicals, NB600-936; diluted 1:200). On occasions when the Total Protein Detection Module (ProteinSimple, DM-TP01) was used for some capillaries, plates were prepared according to the module instructions and were run using the default Total Protein protocol in the Compass software.

## Supporting information

**S1 Fig. Further information on selected *Prnp* guide RNAs. (A**) The top 10 gRNA spacers within codons 23–50 of murine *Prnp* that were identified by the gRNA design algorithm previously found at http://crispr.mit.edu. The scores reported by the algorithm derive from a combination of the on-target efficiency rating and the number of potential off-target sites. **(B)** The chosen *Prnp* gRNA spacer sequences are shown aligned to the murine *Prnp* ORF (NCBI Accession #: NM_011170.3). Due to the preference of the U6 promoter for an initiating G nucleotide, a deliberately mismatched initial G nucleotide was included for gRNA–2 and gRNA–3. **(C)** The highest-scoring predicted cleavages for the second spacer listed in panel A (contained within gRNA–3). Data derives from the same gRNA design algorithm. Sanger sequencing of a PCR amplicon containing the potential off-target site was attempted for each of the six shown. Red highlighting indicates that suitable primers could be designed and that readable sequencing data was obtained.
(TIF)

**S2 Fig. FACS of RK13 cells transiently expressing Cas9 and *Prnp* gRNAs.** WT-5 RK13 cells underwent FACS 48 h after transfection with eSpCas9(1.1) expression plasmids containing *Prnp* gRNA–1, –2 or –3, or no gRNA. Example data for gRNA–1 is shown here, but the same process was applied to each sample. Tight gating based on the side scatter area (SSC-A) vs forward scatter area (FSC-A) plot **(A)** was used to exclude dead cells. **(B, C)** Gates were applied based on SSC and FSC width (W) vs height (H) plots to exclude cell clumps. **(D)** A gate was applied to select the ~10% of cells with the lowest PrP$^C$ signals (P4). **(E)** Summary table showing the cell counts in each population.
(TIF)

**S3 Fig. Checking junctions for correct insertion of the reporter transgene into the N2a cell genome. (A)** Diagnostic digests of pCR2.1–TOPO containing junction PCR products (purified from the gel shown in Fig 3C) were prepared using the indicated restriction enzymes. The expected band sizes were 4.0 and 1.3 kb or 5.1 and 0.25 kb for the 3' junction PCR products (depending on the orientation in which the PCR product inserted into pCR2.1–TOPO) and 3.9 and 1.2 kb for the 5' junction products (irrespective of orientation in pCR2.1–TOPO). All of the 5' junction samples and 5/6 of the 3' junction samples produced the expected band patterns. **(B, C)** Example Sanger sequencing chromatograms for minipreps #1 and #7 showing that the sequences obtained matched the expected junction PCR products, indicating the

absence of unwanted indels around the junction sites. For the expected sequences, underlined letters correspond to the *Gfp-GPI* transgene sequence and non-underlined capitals to the *Prnp* homology arms.
(PDF)

**S4 Fig. Sequencing data from a mouse embryo with CRISPR/Cas9-induced deletions within** ***Prnp.*** **(A)** Sanger sequencing chromatogram for embryo 20 derived from fertilized FVB/NJ oocytes that had been electroporated with recCas9/*Prnp* gRNA–3 RNP complexes. The numbers below the chromatogram indicate the position in the sequencing read rather than the *Prnp* ORF. The data suggested that two relatively large deletions were present within one *Prnp* allele. **(B)** To eliminate the mixed sequence, the original sequencing PCR product was ligated into pCR2.1– TOPO and fresh Sanger sequencing data were obtained from a clone suggested to contain the deletion(s) by diagnostic restriction digests. The alignment with the WT *Prnp* ORF sequence confirms that two deletions of 48 and 27 bp were present in one *Prnp* allele of embryo 20. The exact start and end sites of the deletions cannot be determined due to the presence of microhomologies (underlined). **(C)** The predicted changes to the N-terminal amino acid sequence of mature PrP$^C$ in embryo 20 based on the deletions observed within the *Prnp* ORF.
(TIF)

**S5 Fig. Uncropped agarose gel and capillary western images.** The figure panels that the images correspond to are indicated. A red "X" is used to indicate that a lane was omitted from the final figure panel.
(PDF)

**S1 Table. Primer and DNA sequences.** The ends of the gBlock fragment (underlined) were complementary to the pBud.GFP forward (For) and reverse (Rev) primers to facilitate Gibson Assembly; the sequence in lower case corresponds to codons 230–254 of murine *Prnp*. The PrnpHA–For and–Rev primers both start with 6 random nucleotides followed by a *Sal*I site (For) or *Hind*III site (Rev), the protospacer sequence plus the protospacer-adjacent motif for *Prnp* gRNA–3 (see S1 Fig, panel B, for sequence), and, finally, the *Prnp*-specific sequences. To facilitate Gibson Assembly, the GFP–GPI–For1 and–Rev1 primers contained 18 nt 5' sequences complementary to the ends of the vector fragment amplified by the pB–HA–For and–Rev primers. For the junction PCRs, the Prnp–Intron2–For and GFP–GPI–Rev2 primers were used to analyse the 5' junction, GFP–GPI–For2 and Prnp–3UTR–Rev the 3' junction. The M13–For(-20) and M13–Rev primers were provided as part of a TOPO TA cloning kit (Invitrogen, 450641) and were used to sequence the junction PCR products in both directions.
(PDF)

## Acknowledgments

We thank Rania Faidi for molecular cloning assistance; animal staff at the Centre for Prions and Protein Folding Diseases, University of Alberta, for animal care; staff at the Molecular Biology Services Unit of the Department of Biological Sciences, University of Alberta, for Sanger sequencing; Dr. Feng Zhang for gifting the eSpCas9(1.1) plasmid; and Dr. Jan Langeveld for gifting the 12B2 antibody. Transgenesis was performed at the University of Alberta Faculty of Medicine & Dentistry Transgenic Core and flow cytometry experiments were performed at the University of Alberta Faculty of Medicine & Dentistry Flow Cytometry Facility.

## Author Contributions

**Conceptualization:** Andrew R. Castle, David Westaway.

**Formal analysis:** Andrew R. Castle, Serene Wohlgemuth.

**Funding acquisition:** David Westaway.

**Investigation:** Andrew R. Castle, Serene Wohlgemuth, Luis Arce.

**Project administration:** David Westaway.

**Visualization:** Andrew R. Castle, Serene Wohlgemuth, David Westaway.

**Writing – original draft:** Andrew R. Castle.

**Writing – review & editing:** Andrew R. Castle, Serene Wohlgemuth, Luis Arce, David Westaway.

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
