## [Decision Letter · Decision Letter 0]

22 Apr 2022

PONE-D-22-09021Investigating CRISPR/Cas9 gene drive for production of disease-preventing prion gene allelesPLOS ONE

Dear Dr. Westaway,

Thank you for submitting your manuscript to PLOS ONE. After careful consideration, we feel that it has merit but does not fully meet PLOS ONE’s publication criteria as it currently stands. Therefore, we invite you to submit a revised version of the manuscript that addresses the points raised during the review process.

We look forward to receiving your revised manuscript.

Kind regards,

Rodrigo Morales

Academic Editor

PLOS ONE

Journal Requirements:

Reviewers' comments:

Reviewer's Responses to Questions

**Comments to the Author**

1. Is the manuscript technically sound, and do the data support the conclusions?

Reviewer #1: Yes

Reviewer #2: Yes

2. Has the statistical analysis been performed appropriately and rigorously? 

Reviewer #1: N/A

Reviewer #2: N/A

3. Have the authors made all data underlying the findings in their manuscript fully available?

Reviewer #1: Yes

Reviewer #2: Yes

4. Is the manuscript presented in an intelligible fashion and written in standard English?

Reviewer #1: Yes

Reviewer #2: Yes

5. Review Comments to the Author

Reviewer #1: This manuscript from Castle et al. describes an original approach for the elimination of PrPC expression as a tool against the spread of prion diseases. The authors propose to use CRISPR/Cas9-based technology for the achievement of animals null for Prnp gene. After an initial assay in cellular models, the authors move to the animal model (the mouse). The gene drive promoted by this technology would allow its use in the future as a feasible tool against CWD. It is an ambitious work that implies an important quantity of work in the field of CRISPR/Cas9 technology.

While the results are not those ideally pursued by the authors, the work is well designed and the conclusions obtained are supported by the described results. The publication of this manuscript would help for the development of future strategies to solve the limits observed in the present work. Hence, I would suggest the publication of the manuscript in the present form with minor modifications:

Figure 2. The numbering of the lines is not present in the figure and gRNA nomenclature is not identical in the figure (#1, #2, #3) and in the figure legend (-1, -2, -3).

Figure S1 (B). Define clearly the spacer sequence (#1) used in this work.

Reviewer #2: Chronic wasting disease (CWD) is a rapidly spreading prion disease in wild and captive cervids in North America, Scandinavia, and South Korea for which there is currently no effective disease management strategy. Abnormal folding of a cellular prion protein is the hallmark of the disease. Prion protein knockout cells and animals are resistant to prion replication and diseases caused by prions. Gene drive and Crispr/cas9 technologies provide the necessary approaches for efficient and precise knockout the prion genes in vivo and in vitro. In this study the authors designed specific gRNAs targeting mouse prion coding sequences, and verified the effectiveness and specificity of the system in rabbit kidney epithelial (RK13), mule deer brain cells (MDB) and mouse neuroblastoma N2a cells, and also proved that this system could effectively knock out mouse PrP expression in fertilized mouse eggs using electroporation, unfortunately, the efficiency in male germline was not satisfactory. This study has important scientific significance and practical value to explore the use the gene drive and Crispr/Cas9 to generate CWD resistant cervids.

Comments

1. The authors designed the gRNAs based on mouse Prnp sequence, and tested their efficiency and specificity in mouse N2a cells expressing wild type mouse PrP, and rabbit RK13 cells and mule deer brain MDB cells both engineered to express mouse PrP gene. Could the authors clarify if the gRNAs also target rabbit and mule deer Prnp?

2. In both RK13 and MDB cells, the experiments did not achieve satisfactory results, for MDB cells, it was because the low transfection efficiency; in RK13 cells, the reason for the suboptimal results was due to low levels of Cas9 cleavage. Could the authors discuss what strategies could be used in the future to improve the research?

3. It is known that some PrP gene mutations could lead to spontaneous prion disease. How can the researchers achieve complete elimination of PrP gene expression without causing pathological gene alterations of PrP in animals?

4. Could the authors please update if these gene mutant mouse lines (Lines 33, 34 and 36) could develop normally? Are there any abnormal phenotypes during aging?

6. PLOS authors have the option to publish the peer review history of their article (what does this mean?). If published, this will include your full peer review and any attached files.

Reviewer #1: No

Reviewer #2: No

---

## [Decision Letter · Decision Letter 1]

19 May 2022

Investigating CRISPR/Cas9 gene drive for production of disease-preventing prion gene alleles

PONE-D-22-09021R1

Dear Dr. Westaway,

We’re pleased to inform you that your manuscript has been judged scientifically suitable for publication and will be formally accepted for publication once it meets all outstanding technical requirements.

Kind regards,

Rodrigo Morales

Academic Editor

PLOS ONE

Additional Editor Comments (optional):

Reviewers' comments:

Reviewer's Responses to Questions

**Comments to the Author**

1. If the authors have adequately addressed your comments raised in a previous round of review and you feel that this manuscript is now acceptable for publication, you may indicate that here to bypass the “Comments to the Author” section, enter your conflict of interest statement in the “Confidential to Editor” section, and submit your "Accept" recommendation.

Reviewer #1: All comments have been addressed

Reviewer #2: All comments have been addressed

2. Is the manuscript technically sound, and do the data support the conclusions?

Reviewer #1: Yes

Reviewer #2: Yes

3. Has the statistical analysis been performed appropriately and rigorously? 

Reviewer #1: N/A

Reviewer #2: N/A

4. Have the authors made all data underlying the findings in their manuscript fully available?

Reviewer #1: Yes

Reviewer #2: Yes

5. Is the manuscript presented in an intelligible fashion and written in standard English?

Reviewer #1: Yes

Reviewer #2: Yes

6. Review Comments to the Author

Reviewer #1: The authors have addressed my comments. The article is ready for publication in the current version.

Reviewer #2: (No Response)

7. PLOS authors have the option to publish the peer review history of their article (what does this mean?). If published, this will include your full peer review and any attached files.

Reviewer #1: No

Reviewer #2: No

---

## [Editor Report · Acceptance letter]

30 May 2022

PONE-D-22-09021R1 

Investigating CRISPR/Cas9 gene drive for production of disease-preventing prion gene alleles 

Dear Dr. Westaway:

I'm pleased to inform you that your manuscript has been deemed suitable for publication in PLOS ONE. Congratulations! Your manuscript is now with our production department. 

Kind regards, 

on behalf of

Dr. Rodrigo Morales 

Academic Editor

PLOS ONE